# The Presence of Risk and Protective HLA-DQ Haplotype Combinations and PLA2R1 Risk SNP in Hungarian Patients with Membranous Nephropathy

**DOI:** 10.3390/ijms26178621

**Published:** 2025-09-04

**Authors:** Dóra Bajcsi, Zoltán Maróti, Emőke Endreffy, Péter Légrády, György Ábrahám, Béla Iványi

**Affiliations:** 1Department of Internal Medicine, Albert Szent-Györgyi Medical Centre, Albert Szent-Györgyi Medical School, University of Szeged, 6725 Szeged, Hungary; 2Department of Pediatrics, Albert Szent-Györgyi Medical Centre, Albert Szent-Györgyi Medical School, University of Szeged, 6720 Szeged, Hungary; 3Department of Pathology, Albert Szent-Györgyi Medical Centre, Albert Szent-Györgyi Medical School, University of Szeged, 6725 Szeged, Hungary; ivanyi.bela@med.u-szeged.hu

**Keywords:** primary membranous nephropathy, secondary membranous nephropathy, HLA-DQA/DQB haplotypes, PLA2R1, risk haplotype, protective haplotype

## Abstract

With primary membranous nephropathy (pMN), the genetic background is not precisely known. Certain HLA-DQ serotypes however like HLA-DQ 2.5, and single-nucleotide polymorphisms (SNPs) in the phospholipase A2 receptor 1 (PLA2R1) gene pose a risk for the development of pMN. As antigen presentation is linked to a 3-dimensional conformation of the HLA-DQA/DQB dimer, we thought that the specific HLA-DQ haplotype combinations might also be risk factors in the evolution of MN. The HLA-DQ haplotype combinations and the PLA2R1 gene risk variant (rs4664308) genotypes were examined in 67 patients with MN (52 primary, 15 secondary [sMN]) and 77 controls. Based on the presence or absence of PLA2R1 risk alleles, we used a scoring system to assess the risk and to identify protective HLA-DQ haplotype combinations. The HLA-DQ 2.5 serotype was significantly enriched in both pMN and sMN patients compared to the controls. The pMN group had a significantly higher frequency of the PLA2R1 risk allele compared to the sMN group and the controls. HLA-DQ 2.5 appeared to carry the highest risk for the development of pMN, while HLA-DQ 7.5 and 6.2 seemed to be protective. Our results indicate that the HLA-DQ 2.5 probably carries the highest risk in both pMN and sMN, suggesting that this serotype has less specificity for antigens, and it induces an autoimmune response. Here, PLA2R1 played a role in the development of pMN but not in sMN.

## 1. Introduction

Membranous nephropathy (MN) is a rare immune-complex mediated glomerular disease. Its incidence is about 8 to 12 cases per 1 million worldwide [1]. MN is the main cause of nephrotic syndrome in Caucasian adults [2]. The majority of patients were generally between 50 and 60 years at the time of diagnosis [3,4]. The morphological lesion is characterized by thickened glomerular capillaries with basement membrane spikes along the glomerular basement membrane noted with light microscopy, having a granular staining pattern for IgG and complement 3 along the glomerular capillary loops on immunofluorescence, and subepithelial electron dense deposits on the subepithelial side of the glomerular basement membrane observed with electron microscopy [5]. MN is classified clinically as primary (pMN; 70–80%) or secondary (sMN; 20–30%). Causative factors, such as systemic lupus erythematosus, infections, malignancies and drug exposure cannot be demonstrated in the background of pMN, whereas sMN develops with these in the background [3]. In about 70% of cases with pMN, the immune complexes are composed of circulating autoantibodies, mostly belonging to the IgG4 subclass that react with endogenously expressed podocyte antigen phospholipase A2 receptor 1 (PLA2R1). Several other target antigens have been identified recently in cases of PLA2R-negative MN [5], and the clinical classification of MN is on the way to being replaced by an antigen-based classification.

The genetic background of pMN is currently not entirely clear. Numerous experiments have been performed to link the susceptibility of pMN to different HLA class II gene loci coding proteins essential for the presentation of antigens to T-cells. A significant increase in the HLA-DQA1*0501, DQB1*0301, and DQB1*0601 alleles was noted in Japanese patients with pMN, while the frequency of DQB1*0501 was significantly low [6]. The role of DRB1*0301, DQA1*0501, DQB1*0201 alleles, and therefore HLA-DQ 2.5 serotype (DQA1*0501, DQB1*0201/0202) was also suggested in a British and Greek cohort [7]. In 2011, a multicenter genome-wide association study of single-nucleotide polymorphisms (GWAS SNP) identified two SNPs (rs2187668—HLA-DQA1, rs4664308—PLA2R1) strongly associated with pMN in a large cohort of French, Dutch, and British patients [8]. The SNP rs4664308 is localized in the intronic region of the PLA2R1 gene. The authors came up with a model consisting of a trigger (the immune system), a bullet (PLA2R1 autoantibodies), and a target (glomerular antigen): SNP rs4664308 of the PLA2R1 allele is believed to cause an altered protein conformation, which is presented to the HLA class II receptor and it initiates an autoimmune reaction and in turn, pMN [8].

The role of these two SNPs was also confirmed in a large cohort of non-Caucasian pMN (*n* = 1112) and control patients (*n* = 1020) as well [9]. In a subset of Chinese patients (*n* = 71), the authors showed that out of the patients carrying both risk SNPs, 73% had circulating anti-PLA2R antibodies in the serum and 75% expressed PLA2R1 in glomeruli. They found the low expression of anti-PLA2R antibodies in controls and in patients with hepatitis B-induced sMN; and circulating anti-PLA2R antibodies displayed a high correlation with PLA2R1 expression. They also identified two SNPs that induced a protein alteration in PLA2R1 (rs3749117, M292V and rs35771982, H300D) that correlated (*r*^2^>0.80) with the PLA2R1 risk locus rs4664308 [9]. These studies suggest that these functional SNPs in tight linkage with the previously identified intronic PLA2R1 SNPs are the likely cause of pMN [8,9].

Several studies pin-pointed risk alleles in commonly inherited HLA alleles: DRB1*1501, DRB1*0301, and DRB3*0202 in the Chinese population [10,11], and DRB1*1501 and DQB1*0602 in the Japanese population [12]. In 2020, in a GWAS study of 3782 pMN cases and 9038 controls of East Asian and European descent, three classical risk alleles were also described: DRB1*1501 in East Asians, DQA1*0501 in Europeans (OR = 2.88, *p* = 5.7 × 10^−93^), and DRB1*0301 in both ethnicities [13]. Two previously unreported loci, *NFKB1* (nuclear factor kappa B1; rs230540) and *IRF4* (interferon regulatory factor 4; rs9405192), were also discovered. This association at the *NFKB1* locus has become the focus for ascertaining the role of the NF-κB pathway in pMN [13].

At the DNA level, the HLA-DQA and DQB genes are tightly linked, and in most cases specific HLA-DQA and DQB alleles (serotypes) are inherited together on the same haplotype. However, at the protein level, the HLA-DQA and DQB proteins from the various alleles can freely combine to form the HLA class II protein dimer. Hence, the different allele combinations may result in multiple HLA dimer proteins. The HLA class II antigen-presenting groove is determined by a 3-dimensional conformation of the particular HLA alpha and beta chains of the protein dimer. This is why the potential serotype combinations existing in an individual can only be assessed by the evaluation of the two HLA-DQA/DQB haplotypes.

Here, we wanted to see whether there is a difference between allele counts of HLA-DQ 2.5 haplotype and PLA2R1 risk SNP (rs4664308) in Hungarian patients with pMN and sMN, and in the controls. We also used the tightly linked HLA-DQA1 and DQB1 loci to identify risk and protective haplotype combinations in pMN based on the homozygous or heterozygous state of PLA2R1 risk SNP. We also examined the presence of PLA2R1 risk SNP relative to PLA2R1 histological expression and anti-PLA2R1 antibody serological results.

## 2. Results

### 2.1. PLA2R1 Immunostaining, Serum Anti-PLA2R Antibody Level

Among patients with pMN, PLA2R1 immunostaining positivity was observed in 24 patients (80%). The immunostaining was negative in 6 patients (20%). Regarding the anti-PLA2R antibody serology testing, 13 (76.5%) patients had positive and 4 patients (23.5%) had negative results. In patients with sMN, the PLA2R1 immunostaining and anti-PLA2R immunoserology findings were negative.

### 2.2. PLA2R1 Immunostaining, Anti-PLA2R Antibody Serology for Risk PLA2R1 rs4664308 Alleles

Twenty-four patients were homozygotes for risk PLA2R1 rs4664308 alleles (AA). Nineteen patients had either PLA2R1 immunostaining positivity or serum anti-PLA2R1 antibody positivity, and clinically they were classified as having pMN. Five patients had neither PLA2R1 immunostaining positivity nor serum anti-PLA2R1 antibody positivity; 2 of these 5 patients had clinically diagnosed sMN (see Figure 1).

From the 7 sMN, 4 were heterozygotes for risk PLA2R1 rs4664308 alleles (AG); they had neither positive PLA2R1 immunostaining nor anti-PLA2R1 serology. Of the three patients not carrying PLA2R1 rs4664308 risk alleles (GG), all had no PLA2R1 immunostaining or anti-PLA2R1 immunoserology positivity. Two of them were classified as having sMN, and one of them as having pMN (see Figure 1).

### 2.3. Allele Counts of PLA2R1 rs4664308 SNP and HLA-DQ 2.5 Haplotype

Since both the HLA-DQ 2.5 allele and the PLA2R1 risk SNP were statistically associated with pMN in the literature [7,8,9,10,11], we decided to count the risk and nonrisk alleles for both HLA-DQ 2.5 and the PLA2R1 risk SNP in the different groups. Table 1 gives the allele counts of HLA-DQ2.5 haplotype and PLA2R1 rs4664308 SNP in patients with pMN, sMN, and the control group. The HLA-DQ 2.5 haplotype counts were significantly higher in patients with pMN and sMN than those in the controls; and the difference between the counts of pMN and sMN patients was not significant. The PLA2R1 risk SNP counts were significantly higher in patients with pMN than those in the controls. No significant difference was found between the allele counts of the sMN and control groups. After the Bonferroni correction for multiple hypothesis testing, there was no statistically significant (the modified threshold being 0.016) difference between patients with sMN and pMN (*p* = 0.034), which was most likely due to the lower number of sMN cases (see Table 1).

### 2.4. Scoring Procedure Used for Assessing the HLA-DQ Serotype Combinations in pMN

As described in the Methods section, we created a scoring procedure for the HLA-DQ serotype combinations based on the observed absence or presence of heterozygous or homozygous PLA2R1 risk allele(s) and the disease status of the individual. We arranged the observed HLA-DQ serotype combinations in our cohort based on this scoring system (see Figure 2) to assess the risk and protective serotype combinations. Among the combinations, HLA-DQ 2.5 seemed to carry the highest risk of developing pMN (present in both the first two combinations of the highest score), HLA-DQ 8.1 seemed to carry a risk (present twice in the combinations of the highest scores). As for protective HLA-DQ serotype combinations, HLA-DQ 7.5 and 6.2 seemed to be protective, being present in the most protective combinations (not found in patients with pMN, but occurring in controls carrying the PLA2R1 risk allele).

## 3. Discussion

MN is an umbrella disease that can be classified clinically into primary and secondary types. Although in both types, autoimmune response and specific HLA types were shown to pose an increased risk of developing the disease, the genetic background of MN is still not fully understood. It was found that the presence of the PLA2R1 (chromosome 2) risk SNP (rs4664308) increased the genetic risk for developing pMN, the odds ratio (OR) in a homozygous state being 4.2. However, the HLA-DQA1 risk SNP (rs2187668) in the homozygous state has an even bigger risk, with an OR of 20.2. Combining these two risk alleles with the homozygous state, the OR is 78.5 [8].

After sequencing 30 PLA2R1 coding exons in 95 patients with MN, Coenen et al. could not find any rare genetic variants [14]. This suggests that conformational changes in PLA2R1 probably do not trigger an autoimmune response [3]. One reason for the lack of rare exonic variants might be that the intronic regulatory regions are involved, and post-translational modifications or the increased expression of PLA2R1 antigens may have a role [3]. Another possibility is that despite pMN being a rare disease, common variants in the PLA2R1 gene, combined with common variants of HLA haplotypes, might create a rare haplotype [3,15].

Primary MN can also develop in the absence of the rs4664308 PLA2R1 risk allele as other PLA2R1 variants (rs35771982, rs3749117, rs6757188, rs35771982, rs3828323, rs3749119, rs1511223, rs2203053, rs10196882, rs16844706, rs877635, rs2715928, rs16844715, rs3749119) were also found to play a role in the pathogenesis [9,12,16,17,18]. These findings suggest that the intronic rs4664308 risk allele is only tightly linked to the functional variants that cause the illegitimate expression of PLA2R1 in the kidney.

Among our pMN cases, 76.5% had positive anti-PLA2R antibody serology, and 80% had PLA2R1 immunostaining positivity. In our pMN cohort, 85% of patients carrying two risk PLA2R1 SNPs (AA) had circulating anti-PLA2R1 antibody or expressed PLA2R1 in glomeruli, which is similar that found in Chinese patients [9]. This is why in Hungary, as in other countries, the anti-PLA2R-based pathomechanism is also the main cause of pMN. In sMN cases, these findings were negative, which accords with previous published results [19].

Due to the hypervariability at the MHC II loci, there is a plethora of gene variations (subtypes) of the HLA-DQA1/B1 genes. While these two genes are usually inherited together on tightly linked haplotypes at the DNA level, the DQA1 proteins may freely combine with the DQB1 proteins transcribed from either the cis or trans alleles to form the MHC II dimer protein. This leads to 1 to 4 different HLA-DQ protein dimers/serotypes depending on the two HLA-DQ haplotypes. As the different DQA1 and DQB1 gene variations have slightly different amino acid sequences and protein conformations, the antigen recognition site of the dimer protein is also slightly different based on the actual HLA-DQ serotype. Some haplotypes may also have risk and protective scores depending on the resulting serotype(s), as different combinations can lead to different 3D conformations with different autoimmune responses to PLA2R1.

Based on the assumption that the pathogenesis of pMN is driven by the autoimmune response of specific HLA-DQ serotypes to the illegitimate expression of PLA2R1 in the kidney, we investigated the HLA-DQA1/B1 haplotypes together with the PLA2R1 risk allele (rs4664308) status. We sought to pinpoint the HLA-DQ haplotype combinations that are enriched in MN patients. We also tried to identify those protective HLA-DQ haplotype combinations that are only found in age-matched control patients that carry the PLA2R1 risk alleles without the manifestation of MN.

Since the number of observed haplotype combinations was comparable to the number of sMN cases, we only performed a statistical analysis for the more frequent HLA-DQ 2.5 haplotype allele counts in the three groups (pMN, sMN, controls) that were previously associated with an elevated risk of pMN. We compared patients with pMN and sMN based on HLA-DQ 2.5 haplotype and PLA2R1 risk SNP counts. We noted a significant difference in the counts of HLA-DQ 2.5 between pMN and controls; however, there was no significant difference between pMN and sMN cases (see Table 1), which means that HLA-DQ 2.5 seems to be a strong “lock” in the pathogenesis of MN independently of the “key” used (the immunogenic epitope).

The allele counts of the PLA2R1 risk SNP (rs4664308) were significantly higher in the pMN group compared to the controls. The sMN and control groups had similar risk allele frequencies, and the Fisher exact test was not significant (*p* = 0.686). The pMN group had higher risk allele frequencies compared to the sMN groups; however, after the Bonferroni correction, the significance (*p* = 0.034) fell below the adjusted (*p* = 0.016) threshold (see Table 1). As the relative ratio of risk and non-risk alleles in sMN is comparable to the control, we suggest that this is most likely attributable to the low number of sMN cases. These results tell us that in about 70% of cases, pMN is based on an anti-PLA2R1 antibody-mediated pathomechanism, consistent with findings stated in the literature [9]. These results also suggest that for pMN, the PLA2R1 and the HLA susceptibility factors play a permissive role in the pathogenesis of the disease.

Our findings also confirmed that HLA-DQ 2.5 by itself or combined with some other HLA-DQ haplotypes are overrepresented in patients with pMN compared to controls (see Figure 2), which appear to agree with the published results [7,10,11]. In sMN, the HLA-DQ 2.5 haplotype also appears to carry the highest risk. Recalling that it is a risk haplotype in other autoimmune diseases, such as celiac disease [20,21], it may actually be a general risk trigger of autoimmune response to various epitopes.

MHC II plays a role in antigen presentation; thus, HLA genes play a key role in autoimmune disorders. In the celiac disease HLA-DQ 2.5 heterodimers—encoded by DQA1*0501 and DQB1*0201/0202 alleles both in cis or trans configuration—and DQ8 molecules—encoded by DQB1*03:02 usually in combination with DQA1*03 variant—are known to be genetic susceptibility factors. HLA-DQ 2.5—being present in more than 90% of celiac patients—is more common than HLA-DQ8 [20,21].

Besides the celiac disease, other autoimmune disorders have similar susceptibility HLA alleles to MN: a significant increase was noted in HLA-DRB1*03:01 and *15:01 alleles in systemic lupus erythematosus and in multiple sclerosis [22,23,24,25,26,27]. Furthermore, the HLA-DQ 2.5 haplotype is also associated with other autoimmune diseases, such as type 1 diabetes [28,29], autoimmune hepatitis [30,31], and dermatitis herpetiformis [32]; and HLA-DQ 2.5 has been investigated in rheumatoid arthritis [33,34], and systemic lupus erythematosus [35], although the associations are less clearcut.

Due to the complex interactions and large number of potential serotype combinations that may be present in a patient, the risk assessment of the different HLA-DQ serotypes would require a large dataset. The number of cases in our study does not allow the statistical analysis of the risk or protective HLA-DQ isoforms. However, evaluating the different combinations of potential HLA-DQ haplotypes in patients with MN and controls could help to identify the main risk and protective haplotype combinations. With our biological assumption, we created a risk scoring method to assess the potential role of the HLA-DQ haplotypes. Our scoring system reflects the enrichment or depletion of the PLA2R1 risk alleles observed in the subset of patients with pMN compared to the control persons carrying the particular HLA-DQ haplotype combination. As described in the Methods section, a positive score indicates an increased risk of developing pMN in the presence of the PLA2R1 risk allele, while a negative score suggests a protective role of the given HLA-DQ haplotype combination. Our procedure confirmed that the HLA-DQ 2.5 haplotype has the highest risk and suggests that the HLA-DQ 8.1 haplotype most likely carries an increased risk in developing pMN (see Figure 2). Our approach confirmed the main HLA-DQ haplotype combination already associated with MN and suggested additional risk and potentially protective HLA-DQ haplotypes. Our statistical analysis was limited by the small sample size and the diversity of the HLA-DQ combinations. We think that the same approach could be applied to a bigger cohort to achieve a more precise identification of risk/protective haplotype combinations and proper risk assessment. While our scoring procedure was used for the assessment of pMN risk, a similar approach could be used to analyze the risk and protective HLA-DQ haplotypes in sMN with a suitably large control and sMN dataset, and where HLA-DQ genotype datasets are available.

The findings presented here are in line with those in the current literature. In both pMN and sMN, the main risk haplotype was HLA-DQ 2.5, which seems to be a common “lock” in the pathogenesis of MN, and in the pathogenesis of other autoimmune diseases as well (see Table 1). The presence of PLA2R1 functional SNPs that induce the illegitimate expression of PLA2R1 in the glomeruli is a triggering factor that is required for anti-PLA2R antibody-associated pMN. In the pathogenesis of MN, HLA-DQ haplotype combinations may only be permissive factors, since HLA-DQ susceptibility factor DQ 2.5 haplotype counts are similar in both pMN and sMN groups, and MN will not occur in the presence of protective HLA-DQ haplotype combinations. In sMN, the HLA-DQ susceptibility factors may be the same or different, depending on the various epitopes. However, as the pathogenic antigens (“keys”) of sMN differ (especially in patients with sMN who had lupus nephritis), besides the illegitimate PLA2R1 expression or other antigens of pMN, there may also be differences in the HLA susceptibility factors.

As autoimmune diseases have a complex pathogenesis, apart from genetic factors, environmental and immunological factors are also essential: environmental triggers such as infections, toxins, stress, and diet can induce the disease onset or progression; furthermore, dysregulation of the immune system—including loss of self-tolerance and abnormal activation of immune cells—contributes to the development of autoimmunity. This multifactorial nature makes autoimmune diseases heterogeneous and often difficult to predict or treat. This is why, in vitro or in vivo studies are needed to assess the functional impact of specific HLA-DQ haplotype combinations on antigen presentation and immune response; and experimental confirmation of the biological relevance of these haplotype combinations would strengthen the conclusions drawn above.

These findings may have some potential clinical utility in the future. The identification of HLA-DQ haplotype combinations associated with increased susceptibility to MN could support risk stratification and early detection in genetically predisposed individuals—even among healthy individuals and among patients having an underlying disease predisposed for sMN—such as SLE. And, as our understanding of the immunogenetic landscape improves, therapies targeting specific immune pathways associated with these haplotypes might be mapped out. In individuals carrying genetic risk factors for autoimmune diseases, the use of certain immunomodulatory interventions (for example, vitamin D) may offer a preventive or modulatory effect.

## 4. Patients and Methods

### 4.1. Patients

A total of 67 patients with MN (27 males, 40 females) were analyzed. 22 patients were enrolled prospectively, and 55 patients were enrolled retrospectively. The demographic data of patients and controls are shown in Table 2. Our clinical evaluation (see Table 3) placed 52 patients (77.6%) in the pMN group, and 15 patients (22.4%) in the sMN group. Figure 3 provides a detailed overview of the patient inclusion process. The most common cause of sMN here was lupus nephritis (Table 4).

### 4.2. Histological Evaluation

MN was diagnosed via a morphological evaluation of kidney samples (light microscopic stainings on formalin-fixed and paraffin-embedded tissue sections, immunofluorescence on frozen sections, and electron microscopy) obtained by an ultrasound-guided percutaneous biopsy procedure. Glomerular PLA2R1 immunostaining (indirect IF method, Sigma-Aldrich, Buchs, Switzerland; primary antibody dilution 1:10) was evaluated in 38 patients (31 pMN, 7 sMN); read as negative or positive [4].

### 4.3. Anti-PLA2R Serology

An indirect immunofluorescence semiquantitative assay was used to detect circulating anti-PLA2R antibodies (positivity was assessed at a serum dilution of 1:10) [15]. Blood serum samples were collected for anti-PLA2R serology in 22 patients (17 pMN, 5 sMN) who were in the active stage of MN, in 19 patients at the time of diagnosis, and in 3 patients during a relapse or an active phase with nephrotic syndrome.

Overall, the results of PLA2R1 immunostaining and/or the anti-PLA2R immunoserological analysis were available in 38 patients (30 pMN, 8 sMN).

### 4.4. Genetic Evaluation

A blood sample was taken from the patients with MN and from 77 age-matched, normotensive, clinically healthy persons (Table 2). The persons were enrolled from blood donors of the Regional Blood Bank of Szeged. Real-time polymerase chain reaction (PCR) with a melting curve analysis was used to genotype the rs4664308 SNP in the PLA2R1 gene. HLA-DQA1 and the tightly linked HLA-DQB1 were determined using Inno Lipa diagnostic kits, and the HLA-DQ serotypes were derived from the haplotype results [8].

### 4.5. Scoring of the Risk Property of HLA-DQ Serotype Combinations in pMN

Literature findings suggest that the PLA2R1 risk allele (rs4664308, A) gives rise to the illegitimate expression of PLA2R1 protein [9] in the kidney. The homozygous GG alleles (rs4664308) do not cause the expression of PLA2R1 pathogenic epitopes; in the case of the AG heterozygotes, only a single allele expresses PLA2R1; and in the case of the AA homozygous risk allele combination, there is double illegitimate PLA2R1 expression. In a patient with histologically confirmed MN, a heterozygous PLA2R1 risk allele carries an elevated risk of the HLA-DQ haplotype combination since MN develops even in the case of lower PLA2R1 expression from a single allele. Likewise, the presence of homozygous risk alleles in a control patient suggests a protective HLA-DQ haplotype combination, since even with potentially higher illegitimate PLA2R1 expression, the patient does not develop MN. With this biological assumption, we created a risk scoring system for the observed HLA-DQ serotype combinations. We counted the number of homozygous, heterozygous PLA2R1 rs4664308 risk alleles and the homozygous reference alleles in MN and the control patients. In order to take into account the number of individuals in the different groups, we decided to normalize the count data using allele frequencies based on the sample size.

We marked the allele combinations of the control persons as ‘aa’ (homozygous risk alleles), ‘ag’ (heterozygous risk alleles), and ‘gg’ (homozygous reference alleles). The patients with pMN were assigned in the same way with capital letters (AA: homozygotes to PLA2R1 risk allele; AG: heterozygotes; GG: not carrying the PLA2R1 risk allele). With our biological assumption, we calculated a risk score. For each HLA-DQ serotype, we calculated the frequency of PLA2R1 risk alleles in the pMN and control individuals carrying the particular HLA serotype using the following formula:risk score_HLA type_ = freq (pMN_AA) + 2 × freq (pMN_AG) − 2 × freq (control_AA) − freq (control_AG)

In the case of a neutral effect, we do not expect differences in the allele frequencies of the PLA2R1 risk alleles, and the score should be close to zero. A negative score means a protective effect, while a positive score means an increased risk of the HLA-DQ haplotype combination developing PLA2R1 antigen-based MN.

### 4.6. Statistical Analysis

Fisher’s exact test of the raw count data was used to see whether there was any significant difference in the total number of PLA2R1 risk SNP and HLA-DQ 2.5 haplotype in the groups of pMN, sMN, and control group. If there was any significant difference, as a post-test Fisher’s exact test was used to ascertain whether there was any significant difference in the count for pMN-controls, pMN-sMN, and sMN-controls. Because multiple assumptions were tested, the p threshold was set to 0.05/3 (*p* = 0.017—the Bonferoni correction).

## Figures and Tables

**Figure 1 ijms-26-08621-f001:**
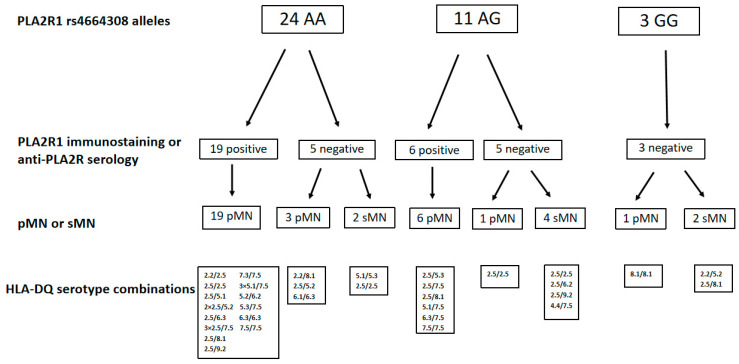
PLA2R1 immunostaining, anti-PLA2R1 antibody serology for risk PLA2R1 rs4664308 alleles. pMN: primary membranous nephropathy, sMN: secondary membranous nephropathy, PLA2R1: phospholipase A2 receptor.

**Figure 2 ijms-26-08621-f002:**
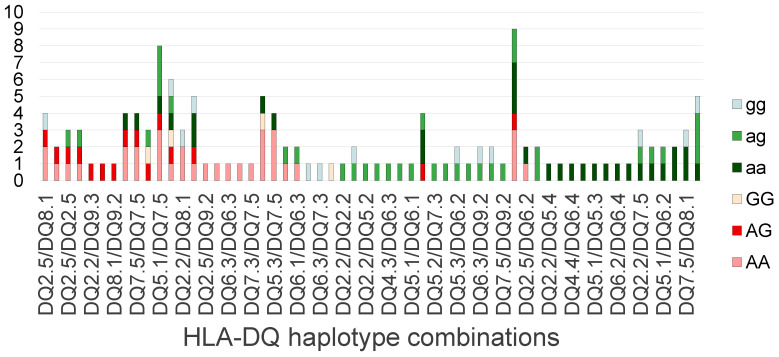
HLA-DQ haplotype combinations in the patients with pMN—risk score (pMN: primary membranous nephropathy).

**Figure 3 ijms-26-08621-f003:**
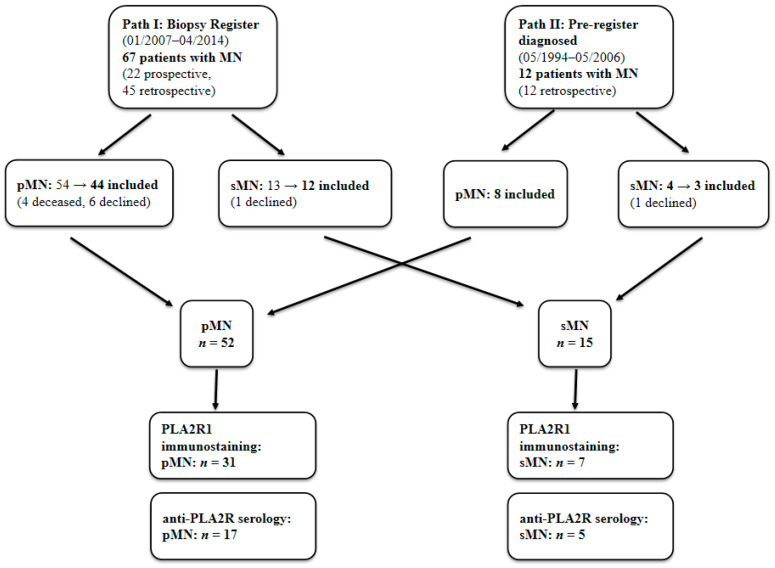
Patient inclusion flow chart. MN: membranous nephropathy, pMN: primary membranous nephropathy, sMN: secondary membranous nephropathy.

**Table 1 ijms-26-08621-t001:** The allele counts for the HLA-DQ 2.5 allele (top) and PLA2R1 risk allele (rs4664308) (bottom).

	Allele Counts		
	pMN	sMN	Control
HLA-DQ 2.5	27 +	11 ++	18
Other HLA-DQ haplotypes	77	19	136
PLA2R1 risk SNP (rs4664308)	81 *	17	94
PLA2R1 non risk SNP	23	13	60

+ pMN is significantly different from the control (*p* = 0.004), but not from sMN (*p* = 0.259). ++ sMN is significantly different from the control (*p* = 0.002). * pMN is significantly different from the control (*p* = 0.005). The difference between pMN and sMN is within the significance threshold after Bonferroni correction (*p* = 0.034). The difference between sMN and the control was not significant (*p* = 0.687). (pMN: primary membranous nephropathy, sMN: secondary membranous nephropathy, PLA2R1: phospholipase A2 receptor, SNP: single-nucleotide polymorphism).

**Table 2 ijms-26-08621-t002:** Demographic data of patients with membranous nephropathy and control persons.

	Membranous Nephropathy	Controls
Number of cases	68 (63 Caucasians, 4 Romas)	77 Caucasians
Age at the time of genetic examinations (years)	50.1 ± 14.7	46.8 ± 8.6
Age at the time of kidney biopsy (years)	46.0 ± 15.6	NA
Female/Male (%)	56/44	65/35

NA: not applicable.

**Table 3 ijms-26-08621-t003:** Protocol used to assign patients to primary membranous nephropathy.

1. Exclusion of underlying autoimmune disorder (clinical signs, laboratory parameters, autoimmune serological tests)
2. Exclusion of viral hepatitis (HbsAg, anti-HCV)
3. Exclusion of malignancy (laboratory parameters, chest X-ray, abdominal and pelvic ultrasound, gynecological and urological examinations, fecal occult blood test; in patients over 60 years of age: gastroscopy, colonoscopy)
4. Exploration of medications potentially causing drug-induced MN (NSAIDs, gold, penicillamine)
5. No manifestation of autoimmune disorders or malignancy within two years after the kidney biopsy evaluation

MN: membranous nephropathy, HbsAg: hepatitis B surface antigen, HCV: hepatitis C virus, NSAIDs: nonsteroidal anti-inflammatory drugs.

**Table 4 ijms-26-08621-t004:** Types of secondary membranous nephropathy cases.

Types of sMN	Number of Cases
**Systemic autoimmune diseases**	
Membranous lupus nephritis	8
Mixed tissue connective disease	1
Rheumatoid arthritis (+ in 1 case: gold exposure)	2
**Organ-specific autoimmune diseases**	
Hashimoto thyroiditis	1
Autoimmune hepatitis	1
**Viral hepatitis**	
Hepatitis B (+ Graves-Basedow disease)	1
Hepatitis C (+ lupus nephritis)	1

sMN: secondary membranous nephropathy.

## Data Availability

The datasets used and/or analyzed during the current study are available from the corresponding author upon reasonable request.

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
