# Peer review of "The Presence of Risk and Protective HLA-DQ Haplotype Combinations and PLA2R1 Risk SNP in Hungarian Patients with Membranous Nephropathy"

_ijms, 2025, doi:10.3390/ijms26178621_

Round 1
Reviewer 1 Report
Comments and Suggestions for Authors
This study investigates the genetic background of membranous nephropathy (MN), focusing on Hungarian patients. The authors analyze the association between specific HLA-DQ haplotype combinations and the PLA2R1 risk single nucleotide polymorphism (SNP) rs4664308 in both primary (pMN) and secondary (sMN) MN cases. Their results demonstrate that the HLA-DQ 2.5 serotype is significantly enriched in both pMN and sMN compared to controls, while the PLA2R1 risk allele is more frequent in pMN. The study introduces a risk scoring system that combines HLA-DQ haplotype and PLA2R1 genotype data, identifying HLA-DQ 2.5 as the highest risk haplotype for MN, with HLA-DQ 7.5 and 6.2 showing potential protective effects. These findings contribute to the understanding of MN pathogenesis and suggest that both antigen presentation and genetic susceptibility play critical roles in disease development, which may inform future diagnostic and therapeutic strategies.
One key limitation of the study is the relatively small sample size, particularly in the sMN group, which restricts the statistical power and generalizability of the findings.
Another concern is the lack of functional validation for the proposed risk and protective haplotypes; while the scoring system is innovative, experimental confirmation of the biological relevance of these haplotypes would strengthen the conclusions. The authors are encouraged to incorporate in vitro or in vivo studies to assess the functional impact of specific HLA-DQ haplotype combinations on antigen presentation and immune response.
Additionally, the study does not address potential confounding factors such as population stratification or environmental influences that may affect allele frequencies.
Finally, the discussion could benefit from a deeper exploration of how these genetic findings might translate into clinical practice, such as risk stratification or personalized treatment approaches, to highlight the translational value of the research.
Author Response
Answers to Reviewer 1:
We thank you for the insightful and fruitful remarks. We hope that the proposed revisions and suggestions will enhance the clarity of the manuscript.
We provide a point-by-point response to each of the concerns raised.
The small sample size is a justified argument from the reviewer. We mentioned in the Conclusion part: “Our statistical analysis was limited by the small sample size and the diversity of the HLA-DQ combinations. We think that that the same approach could be applied to a bigger cohort to achieve a more precise identification of risk/protective haplotype combinations and proper risk assessment. While our scoring procedure was used for the assessment of pMN risk, a similar approach could be used to analyze the risk and protective HLA-DQ haplotypes in sMN with a suitably large control and sMN dataset, and where HLA-DQ genotype datasets are available.“
The concern of the lack of functional validation for the proposed risk and protective haplotypes and the potential impact of environmental factors is reasonable. A part of our study was retrospective, which inherently limits the feasibility of performing a functional analysis in hindsight. We think that addressing this limitation would require a prospective investigation in a larger cohort. We fully acknowledge the reviewer’s valid point that, if feasible, a functional analysis could provide concrete evidence supporting the proposed pathomechanistic hypothesis. We revised the Conclusion part to include this aspect as follows:
“As autoimmune diseases have a complex pathogenesis, apart from genetic factors, environmental and immunological factors are also essential: environmental triggers such as infections, toxins, stress, and diet can induce the disease onset or progression; furthermore, dysregulation of the immune system —including loss of self-tolerance and abnormal activation of immune cells— contributes to the development of autoimmunity. This multifactorial nature makes autoimmune diseases heterogeneous and often difficult to predict or treat. This is why, in vitro or in vivo studies are needed to assess the functional impact of specific HLA-DQ haplotype combinations on antigen presentation and immune response; and experimental confirmation of the biological relevance of these haplotype combinations would strengthen the conclusions drawn above.
These findings may have some potential clinical utility in the future. The identification of HLA-DQ haplotype combinations associated with increased susceptibility to MN could support risk stratification and early detection in genetically predisposed individuals – even among healthy individuals and among patients having an underlying disease predisposed for sMN – such as SLE. And, as our understanding of the immunogenetic landscape improves, therapies targeting specific immune pathways associated with these haplotypes might be mapped out. In individuals carrying genetic risk factors for autoimmune diseases, the use of certain immunomodulatory interventions (for example, vitamin D) may offer a preventive or modulatory effect.”
Reviewer 2 Report
Comments and Suggestions for Authors
Thank you for inviting me to review this study. This is a well-written and presented study. Only minor comments that would develop the quality of the paper:
- Please use politically correct language. By this, I mean substituting phrases like MN patients to patients with MN.
- When a sentence starts with numerical references, please don’t use numbers but letters. Additionally, do not use abbreviations as the first word of a sentence.
- A study flow chart of selected patients is needed.
- Could you provide a ref of your prespecified protocol in the methods section?
- Please change “renal” to “kidney” as this is the right reference according to KDIGO.
Author Response
Answers to Reviewer 2:
We thank you the comments, and hope that the proposed revisions and modifications will enhance the clarity of the manuscipt. We provide a point-by-point response to each of the concerns raised.
We corrected:
- MN patients to patients with MN (1.)
- the sentence starting with numerical references to a letter (2.)
- renal to kidney (5.)
We created a patient inclusion flow chart (3.).
We provided a reference for the prespecified protocol in the Method section (4.).
Round 2
Reviewer 1 Report
Comments and Suggestions for Authors
All concerns have been addressed.